# `RobustBench`: a standardized adversarial robustness benchmark

**Francesco Croce**[*]
Univ. of Tübingen

**Maksym Andriushchenko**[*]
EPFL

**Vikash Sehwag**[*]
Princeton Univ.

**Edoardo Debenedetti**[*]
EPFL

**Nicolas Flammarion**
EPFL

**Mung Chiang**
Purdue Univ.

**Prateek Mittal**
Princeton Univ.

**Matthias Hein**
Univ. of Tübingen

## Abstract

As a research community, we are still lacking a systematic understanding of the progress on adversarial robustness which often makes it hard to identify the most promising ideas in training robust models. A key challenge in benchmarking robustness is that its evaluation is often error-prone leading to *robustness overestimation*. Our goal is to establish a *standardized benchmark* of adversarial robustness, which as accurately as possible reflects the robustness of the considered models within a reasonable computational budget. To this end, we start by considering the image classification task and introduce restrictions (possibly loosened in the future) on the allowed models. We evaluate adversarial robustness with *AutoAttack* [28], an ensemble of white- and black-box attacks, which was recently shown in a large-scale study to improve almost all robustness evaluations compared to the original publications. To prevent overadaptation of new defenses to AutoAttack, we welcome external evaluations based on adaptive attacks [142], especially where AutoAttack flags a potential overestimation of robustness. Our leaderboard, hosted at https://robustbench.github.io/, contains evaluations of 120+ models and aims at reflecting the current state of the art in image classification on a set of well-defined tasks in $\ell_\infty$- and $\ell_2$-threat models and on common corruptions, with possible extensions in the future. Additionally, we open-source the library https://github.com/RobustBench/robustbench that provides unified access to 80+ robust models to facilitate their downstream applications. Finally, based on the collected models, we analyze the impact of robustness on the performance on distribution shifts, calibration, out-of-distribution detection, fairness, privacy leakage, smoothness, and transferability.

## 1 Introduction

Since the finding that state-of-the-art deep learning models are vulnerable to small input perturbations called *adversarial examples* [135], achieving adversarially robust models has become one of the most studied topics in the machine learning community. The main difficulty of robustness evaluation is that it is a computationally hard problem even for simple $\ell_p$-bounded perturbations [70] and exact approaches [138] do not scale to large enough models. There are already more than 3000 papers on this topic [16], but it is often unclear which defenses against adversarial examples indeed improve robustness and which only make the typically used attacks overestimate the actual robustness. There is an important line of work on recommendations for how to perform adaptive attacks that are selected specifically for a particular defense [5, 17, 142] which have in turn shown that several

---

[*]Equal contribution.

35th Conference on Neural Information Processing Systems (NeurIPS 2021) Track on Datasets and Benchmarks.

| Rank ▲ | Method | Standard accuracy | AutoAttack robust accuracy | Best known robust accuracy | AA eval. potentially unreliable | Extra data | Architecture | Venue |
|---|---|---|---|---|---|---|---|---|
| 1 | Fixing Data Augmentation to Improve Adversarial Robustness
*66.56% robust accuracy is due to the original evaluation (AutoAttack + MultiTargeted)* | 92.23% | 66.58% | 66.56% | ✗ | ☑ | WideResNet-70-16 | arXiv, Mar 2021 |
| 2 | Uncovering the Limits of Adversarial Training against Norm-Bounded Adversarial Examples
*65.87% robust accuracy is due to the original evaluation (AutoAttack + MultiTargeted)* | 91.10% | 65.88% | 65.87% | ✗ | ☑ | WideResNet-70-16 | arXiv, Oct 2020 |
| 3 | Fixing Data Augmentation to Improve Adversarial Robustness
*It uses additional 1M synthetic images in training. 64.58% robust accuracy is due to the original evaluation (AutoAttack + MultiTargeted)* | 88.50% | 64.64% | 64.58% | ✗ | ✗ | WideResNet-106-16 | arXiv, Mar 2021 |

Figure 1: The top-3 entries of our CIFAR-10 leaderboard hosted at `https://robustbench.github.io/` for the $\ell_\infty$-perturbations of radius $\varepsilon_\infty = 8/255$.

seemingly robust defenses fail to be robust. However, recently Tramèr et al. [142] observe that although several recently published defenses have tried to perform adaptive evaluations, many of them could still be broken by new adaptive attacks. We observe that there are repeating patterns in many of these defenses that prevent standard attacks from succeeding. This motivates us to impose restrictions on the defenses we consider in our proposed benchmark, `RobustBench`, which aims at *standardized* adversarial robustness evaluation. Specifically, we rule out (1) classifiers which have zero gradients with respect to the input [13, 53], (2) randomized classifiers [161, 100], and (3) classifiers that use an optimization loop at inference time [118, 84]. Often, non-certified defenses that violate these three restrictions only make gradient-based attacks harder but do not substantially improve robustness [17]. However, we will lift (some of) these constraints if a standardized reliable evaluation method for those defenses becomes available. We start from benchmarking robustness with respect to the $\ell_\infty$- and $\ell_2$-threat models, since they are the most studied settings in the literature. We use the recent AutoAttack [28] as our current standard evaluation which is an ensemble of diverse parameter-free attacks (white- and black-box) that has shown reliable performance over a large set of models that satisfy our restrictions. Moreover, we accept and encourage external evaluations, e.g. with adaptive attacks, to improve our standardized evaluation, especially for the leaderboard entries whose evaluation may be unreliable according to the *flag* that we propose. Additionally, we collect models robust against common image corruptions [58] as these represent another important type of perturbations which should not change the decision of a classifier.

**Contributions.** We make following key contributions with our `RobustBench` benchmark:

- **Leaderboard** (`https://robustbench.github.io/`): a website with the leaderboard (see Fig. 1) based on *more than 120* evaluations where it is possible to track the progress and the current state of the art in adversarial robustness based on a standardized evaluation using AutoAttack *complemented* by (external) adaptive evaluations. The goal is to clearly identify the most successful ideas in training robust models to accelerate the progress in the field.

- **Model Zoo** (`https://github.com/RobustBench/robustbench`): a collection of the most robust models that are easy to use for any downstream applications. As an example, we expect that this will foster the development of better adversarial attacks by making it easier to perform evaluations on a large set of *more than 80* models.

- **Analysis**: based on the collected models from the Model Zoo, we provide an analysis of how robustness affects the performance on distribution shifts, calibration, out-of-distribution detection, fairness, privacy leakage, smoothness, and transferability. In particular, we find that robust models are significantly *underconfident* that leads to worse calibration, and that not all robust models have higher privacy leakage than standard models.

## 2 Background and related work

**Adversarial perturbations.** Let $x \in \mathbb{R}^d$ be an input point and $y \in \{1, \ldots, C\}$ be its correct label. For a classifier $f : \mathbb{R}^d \to \mathbb{R}^C$, we define a *successful adversarial perturbation* with respect to the

perturbation set $\Delta \subseteq \mathbb{R}^d$ as a vector $\boldsymbol{\delta} \in \mathbb{R}^d$ such that

$$\arg\max_{c \in \{1,...,C\}} f(\boldsymbol{x} + \boldsymbol{\delta})_c \neq y \quad \text{and} \quad \boldsymbol{\delta} \in \Delta, \tag{1}$$

where typically the perturbation set $\Delta$ is chosen such that *all* points in $x + \delta$ have $y$ as their true label. This motivates a typical robustness measure called *robust accuracy*, which is the fraction of datapoints on which the classifier $f$ predicts the correct class for all possible perturbations from the set $\Delta$. Computing the exact robust accuracy is in general intractable and, when considering $\ell_p$-balls as $\Delta$, NP-hard even for single-layer neural networks [70, 149]. In practice, an *upper bound* on the robust accuracy is computed via some *adversarial attacks* which are mostly based on optimizing some differentiable loss (e.g., cross entropy) using local search algorithms like projected gradient descent (PGD) in order to find a successful adversarial perturbation. The tightness of the upper bound depends on the effectiveness of the attack: unsuitable techniques or suboptimal parameters (e.g., the step size and the number of iterations) can make the models appear more robust than they actually are [36, 95], especially in the presence of phenomena like gradient obfuscation [5]. Certified methods [151, 48] instead provide *lower bounds* on robust accuracy which often underestimate robustness significantly, especially if the certification was not part of the training process. Thus, in our benchmark, we do not measure lower bounds and focus only on upper bounds which are typically much tighter [138].

**Threat models.** We focus on the fully white-box setting, i.e. the model $f$ is assumed to be fully known to the attacker. The threat model is defined by the set $\Delta$ of the allowed perturbations: the most widely studied ones are the $\ell_p$-perturbations, i.e. $\Delta_p = \{\boldsymbol{\delta} \in \mathbb{R}^d, \|\boldsymbol{\delta}\|_p \leq \varepsilon\}$, particularly for $p = \infty$ [135, 46, 88]. We rely on thresholds $\varepsilon$ established in the literature which are chosen such that the true label should stay the same for each in-distribution input within the perturbation set. We note that robustness towards small $\ell_p$-perturbations is a necessary but not sufficient notion of robustness which has been criticized in the literature [45]. It is an active area of research to develop threat models which are more aligned with the human perception such as spatial perturbations [42, 39], Wasserstein-bounded perturbations [152, 63], perturbations of the image colors [80] or $\ell_p$-perturbations in the latent space of a neural network [81, 150]. However, despite the simplicity of the $\ell_p$-perturbation model, it has numerous interesting applications that go beyond security considerations [141, 116] and span transfer learning [117, 145], interpretability [143, 71, 38], generalization [158, 174, 9], robustness to unseen perturbations [68, 158, 81, 73], stabilization of GAN training [173]. Thus, improvements in $\ell_p$-robustness have the potential to improve many of these downstream applications.

Additionally, we provide leaderboards for *common image corruptions* [58] that try to mimic modifications of the input images which can occur naturally. Unlike $\ell_p$ adversarial perturbations, they are not imperceptible and evaluation on them is done in the average-case fashion, i.e. there is no attacker who aims at changing the classifier's decision. In this case, the robustness of a model is evaluated as classification accuracy on the corrupted images, averaged over corruption types and severities.

**Related libraries and benchmarks.** There are many libraries that focus primarily on implementations of popular adversarial attacks such as FoolBox [110], Cleverhans [104], AdverTorch [33], AdvBox [47], ART [98], SecML [92], DeepRobust [85]. Some of them also provide implementations of several basic defenses, but they do not include up-to-date state-of-the-art models. The two challenges [79, 10] hosted at NeurIPS 2017 and 2018 aimed at finding the most robust models for specific attacks, but they had a predefined deadline, so they could capture the best defenses only at the time of the competition. Ling et al. [86] proposed DEEPSEC, a benchmark that tests many combinations of attacks and defenses, but suffers from a few shortcomings as suggested by Carlini [15]: (1) reporting average-case instead of worst-case performance over multiple attacks, (2) evaluating robustness in threat models different from the ones used for training, (3) using excessively large perturbations. Chen and Gu [21] proposed a new *hard-label* black-box attack, RayS, and evaluated it on a range of models which led to a leaderboard (https://github.com/uclaml/RayS). Despite being a state-of-the-art hard-label black-box attack, the robust accuracy in the leaderboard given by RayS still tends to be overestimated even compared to the original evaluations.

Recently, Dong et al. [35] have provided an evaluation of a few defenses (in particular, 3 for $\ell_\infty$- and 2 for $\ell_2$-norm on CIFAR-10) against multiple commonly used attacks. However, they did not include some of the best performing defenses [60, 19, 50, 111] and attacks [49, 27], and in a few cases, their evaluation suggests robustness higher than what was reported in the original papers. Moreover, they do not impose any restrictions on the models they accept to the benchmark. RobustML (https://www.robust-ml.org/) aims at collecting robustness claims for defenses together with external evaluations. Their format does not assume running any baseline attack, so it relies entirely

on evaluations submitted by the community, which however do not occur often enough. Thus even though RobustML has been a valuable contribution to the community, now it does not provide a comprehensive overview of the recent state of the art in adversarial robustness.

Finally, it has become common practice to test new attacks wrt $\ell_\infty$ on the publicly available models from Madry et al. [88] and Zhang et al. [168], since those represent widely accepted defenses which have stood many thorough evaluations. However, having only two models per dataset (MNIST and CIFAR-10) does not constitute a sufficiently large testbed, and, because of the repetitive evaluations, some attacks may already overfit to those defenses.

**What is different in** `RobustBench`**.** Learning from these previous attempts, `RobustBench` presents a few different features compared to the aforementioned benchmarks: (1) a baseline worst-case evaluation with an ensemble of *strong*, *standardized* attacks [28] which includes both white- and black-box attacks, unlike RobustML which is *solely* based on adaptive evaluations, integrated by external evaluations, (2) we add a *flag* in AutoAttack raised when the evaluation might be unreliable, in which case we do additional adaptive evaluations ourselves and encourage the community to contribute, (3) clearly defined threat models that correspond to the ones used during training of submitted models, (4) evaluation of not only standard defenses [88, 168] but also of more recent improvements such as [19, 50, 111]. Moreover, `RobustBench` is designed as an *open-ended* benchmark that keeps an up-to-date leaderboard, and we welcome contributions of new defenses and evaluations using adaptive attacks. Finally, we open source the Model Zoo for convenient access to the 80+ most robust models from the literature which can be used for downstream tasks and facilitate the development of new standardized attacks.

## 3   Description of `RobustBench`

We start by providing a detailed layout of our proposed leaderboards for $\ell_\infty$, $\ell_2$, and common corruption threat models. Next, we present the Model Zoo, which provides unified access to most networks from our leaderboards.

### 3.1   Leaderboard

**Restrictions.** We argue that accurate benchmarking adversarial robustness in a standardized way requires some restrictions on the type of considered models. The goal of these restrictions is to prevent submissions of defenses that cause some standard attacks to fail without truly improving robustness. Specifically, we consider only classifiers $f : \mathbb{R}^d \to \mathbb{R}^C$ that

- have in general *non-zero gradients* with respect to the inputs. Models with zero gradients, e.g., that rely on quantization of inputs [13, 53], make gradient-based methods ineffective thus requiring zeroth-order attacks, which do not perform as well as gradient-based attacks. Alternatively, specific adaptive evaluations, e.g. with Backward Pass Differentiable Approximation [5], can be used which, however, can hardly be standardized. Moreover, we are not aware of existing defenses solely based on having zero gradients for large parts of the input space which would achieve competitive robustness.

- have a *fully deterministic forward pass*. To evaluate defenses with stochastic components, it is a common practice to combine standard gradient-based attacks with Expectation over Transformations [5]. While often effective it might be not sufficient, as shown by Tramèr et al. [142]. Moreover, the classification decision of randomized models may vary over different runs for the same input, hence even the definition of robust accuracy differs from that of deterministic networks. We note that randomization *can* be useful for improving robustness and deriving robustness certificates [82, 25], but it also introduces variance in the gradient estimators (both white- and black-box) making standard attacks much less effective.

- do not have an *optimization loop* in the forward pass. This makes backpropagation through it very difficult or extremely expensive. Usually, such defenses [118, 84] need to be evaluated adaptively with attacks that rely on a combination of hand-crafted losses.

Some of these restrictions were also discussed by [12] for the warm-up phase of their challenge. We refer the reader to Appendix E therein for an illustrative example of a trivial defense that bypasses gradient-based and some of the black-box attacks they consider. We believe that such constraints

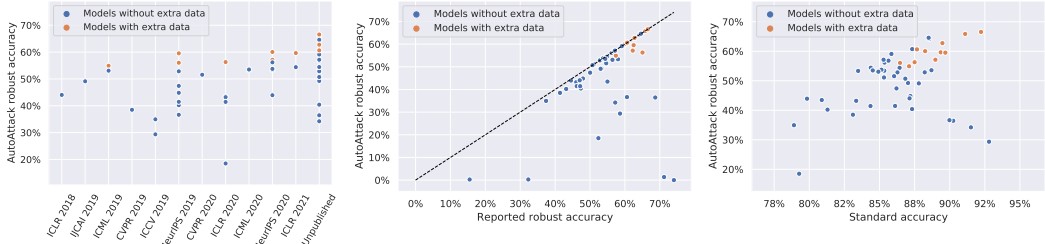

Figure 2: Visualization of the robustness and accuracy of 54 CIFAR-10 models from the `RobustBench` $\ell_\infty$-leaderboard. Robustness is evaluated using $\ell_\infty$-perturbations with $\varepsilon_\infty = 8/255$.

are necessary at the moment since they allow an accurate standardized evaluation which makes the leaderboard meaningful and sustainable. In fact, for defenses not fulfilling the restrictions there is no standard evaluation which is shown to generalize and perform well across techniques, thus one has to resort to time-consuming adaptive attacks specifically tailored for each case. In the design of our benchmark, we thought that it is more important that the robustness evaluation is reliable, rather than being open to all possible defenses with the risk that the robustness is drastically overestimated. As this can lead to a potential bias in our leaderboard, we will lift the restrictions if reliable standardized evaluation methods for these modalities become available in the literature.

**Overall setup.** We set up leaderboards for the $\ell_\infty$, $\ell_2$ and common corruption threat models on CIFAR-10, CIFAR-100 [75], and ImageNet [32] datasets (see Table 1 for details). We use the fixed budgets of $\varepsilon_\infty = 8/255$ and $\varepsilon_2 = 0.5$ for the $\ell_\infty$ and $\ell_2$ leaderboards for CIFAR-10 and CIFAR-100. For ImageNet, we use $\varepsilon_\infty = 4/255$ and in App. D, we discuss how we handle that different models use different image resolutions for ImageNet. Most of the models shown in the leaderboards are taken from papers published at top-tier machine learning and computer vision conferences as shown in Fig. 2 (left). For each entry we report the reference to the original paper, standard and robust accuracy under the specific threat model (see the next paragraph for details), network architecture, venue where the paper appeared and possibly notes regarding the model. We also highlight when extra data (often, the dataset introduced by Carmon et al. [19]) is used since it gives a clear advantage for both clean and robust accuracy. If any other attack achieves lower robust accuracy than AutoAttack then we also report it. Moreover, the leaderboard allows to search the entries by their metadata (such as title, architecture, venue) which can be useful to compare different methods that use the same architecture or to search for papers published at some conference.

**Evaluation of defenses.** The evaluation of robust accuracy on common corruptions [58] involves simply computing the average accuracy on corrupted images over different corruption types and severity levels.[1] To evaluate robustness of $\ell_\infty$ and $\ell_2$ defenses, we currently use AutoAttack [28]. It is an ensemble of four attacks that are run sequentially: a variation of PGD attack with automatically adjusted step sizes, with (1) the cross entropy loss and (2) the difference of logits ratio loss, which is a rescaling-invariant margin-based loss function, (3) the targeted version of the FAB attack [27], which minimizes the $\ell_p$-norm of the perturbations, and (4) the black-box Square Attack [4]. Each subsequent attack is run on the points for which an adversarial example has not been found by the preceding attacks. We choose AutoAttack as it includes both black-box and white-box attacks, does not require hyperparameter tuning (in particular, the step size), and consistently improves the results reported in the original papers for almost all the models (see Fig. 2 (middle)). If in the future some new standardized and parameter-free attack is shown to consistently outperform AutoAttack on a wide set of models given a similar computational cost, we will adopt it as standard evaluation. In order to verify the reproducibility of the results, we perform the standardized evaluation independently of the authors of the submitted models. We encourage evaluations of the models in the leaderboard with adaptive or external attacks to reflect the best available upper bound on the true robust accuracy (see a pre-formatted issue template in our repository[2]), in particular in the case where AutoAttack flags that it might not be reliable (see paragraph below). For example, Gowal et al. [50] and Rebuffi et al. [111] evaluate their models with a hybrid of AutoAttack and MultiTargeted attack [49], that in some cases reports slightly lower robust accuracy than AutoAttack alone. We reflect the additional

---

[1]A breakdown over corruptions and severities is also available, e.g. for CIFAR-10 models see: `https://github.com/RobustBench/robustbench/blob/master/model_info/cifar10/corruptions/unaggregated_results.csv`

[2]`https://github.com/RobustBench/robustbench/issues/new/choose`

evaluations in our leaderboard by reporting in a separate column the robust accuracy for the worst case of AutoAttack and all other evaluations. Below we show an example of how one can use our library to easily benchmark a model (either external one or taken from the Model Zoo):

```
from robustbench.eval import benchmark
clean_acc, robust_acc = benchmark(model, dataset='cifar10', threat_model='Linf')
```

Moreover, in Appendix E we also show the variability of the robust accuracy given by AutoAttack over random seeds and report its runtime for a few models from different threat models.

**Identifying potential need for adaptive attacks.** Although AutoAttack provides an accurate estimation of robustness for most models that satisfy the restrictions mentioned above, there might still be corner cases when AutoAttack overestimates robustness of a model that satisfies the restrictions. Carlini et al. [17] suggest that one indicator of possible overestimation of robustness is when black-box attacks are more effective than white-box ones. We noticed that this is the case for the model from Xiao et al. [156] where the black-box Square Attack [4] improves by *more than* $10\%$ the robust accuracy given by the previous white-box attacks in AutoAttack. We run a simple adaptive attack: Square Attack with multiple random restarts (30 instead of 1) decreases the robust accuracy from the $18.50\%$ of AutoAttack to $7.40\%$. We note that AutoAttack did not fail completely for this model and correctly revealed a lower level of robustness than reported ($52.4\%$), although the exact robust accuracy was overestimated. Based on this case, we integrate a *flag* in AutoAttack: a warning is output whenever Square Attack reduces of more than $0.2\%$ the robust accuracy compared to the white-box gradient-based attacks in AutoAttack. In this case, AutoAttack evaluation might be not fully reliable and adaptive attacks might be necessary, so we flag the corresponding entries in the leaderboard (currently, only the model of Xiao et al. [156]). Moreover, for the sake of convenience, we also integrate in AutoAttack flags that automatically inform the user if the restrictions are violated.[3]

**Adding new defenses.** We believe that the leaderboard is only useful if it reflects the latest advances in the field, so it needs to be constantly updated with new defenses. We intend to include evaluations of new techniques and we welcome contributions from the community which help to keep the benchmark up-to-date. We require new entries to (1) satisfy the three restrictions stated above, (2) to be accompanied by a publicly available paper (e.g., an arXiv preprint) describing the technique used to achieve the reported results, and (3) share the model checkpoints (not necessarily publicly). We also allow *temporarily* adding entries without providing checkpoints given that the authors evaluate their models with AutoAttack. However, we will mark such evaluations as *unverified*, and to encourage reproducibility, we reserve the right to remove an entry later on if the corresponding model checkpoint is not provided. It is possible to add a new defense to the leaderboard and (optionally) the Model Zoo by opening an issue with a predefined template in our repository `https://github.com/RobustBench/robustbench`, where more details about new additions can be found.

### 3.2 Model Zoo

We collect the checkpoints of many networks from the leaderboard in a single repository hosted at `https://github.com/RobustBench/robustbench` after obtaining the permission of the authors (see Appendix B for the information on the licenses). The goal of this repository, the Model Zoo, is to make the usage of robust models as simple as possible to facilitate various downstream applications and analyses of general trends in the field. In fact, even when the checkpoints of the proposed method are made available by the authors, it is often time-consuming and not straightforward to integrate them in the same framework because of many factors such as small variations in the architectures, custom input normalizations, etc. For simplicity of implementation, at the moment we include only models implemented in PyTorch [105]. Below we illustrate how a model can be automatically downloaded and loaded via its identifier and threat model within two lines of code:

```
from robustbench.utils import load_model
model = load_model(model_name='Ding2020MMA', dataset='cifar10', threat_model='L2')
```

At the moment, all models (see Table 1 and Appendix G for details) are variations of ResNet [55] and WideResNet architectures [164] of different depth and width. However, we note that the benchmark and Model Zoo are not restricted only to residual or convolutional networks, and we are ready to

---

[3]See `https://github.com/fra31/auto-attack/blob/master/flags_doc.md` for details

Table 1: The total number of models in the Model Zoo and leaderboards per dataset and threat model.

| Threat model | CIFAR-10 | | CIFAR-100 | | ImageNet | |
|---|---|---|---|---|---|---|
| | Model Zoo | Leaderboard | Model Zoo | Leaderboard | Model Zoo | Leaderboard |
| $\ell_\infty$ | 39 | 63 | 14 | 14 | 5 | 6 |
| $\ell_2$ | 17 | 18 | - | - | - | - |
| Common corruptions [58] | 7 | 15 | 2 | 4 | 5 | 7 |

add any other architecture. We include the most robust models, e.g. those from Rebuffi et al. [111], but there are also defenses which pursue additional goals alongside adversarial robustness at the fixed threshold we use: e.g., Sehwag et al. [122] consider networks which are robust and compact, Wong et al. [153] focus on computationally efficient adversarial training, Ding et al. [34] aim at input-adaptive robustness as opposed to robustness within a single $\ell_p$-radius. All these factors have to be taken into account when comparing different techniques, as they have a strong influence on the final performance. Thus, we highlight these factors in the footnotes below each paper's title.

**A testbed for new attacks.** Another important use case of the Model Zoo is to simplify comparisons between different adversarial attacks on a wide range of models. First, the leaderboard already serves as a strong baseline for new attacks. Second, as mentioned above, new attacks are often evaluated on the models from Madry et al. [88] and Zhang et al. [168], but this may not provide a representative picture of their effectiveness. For example, currently the difference in robust accuracy between the first and second-best attacks in the CIFAR-10 leaderboard of Madry et al. [88] is only 0.03%, and between the second and third is 0.04%. Thus, we believe that a more thorough comparison should involve multiple models to prevent overfitting of the attack to one or two standard robust defenses.

## 4 Analysis

With unified access to multiple models from the Model Zoo, one can easily compute various performance metrics to see general trends. We analyze various aspects of robust classifiers, mostly for $\ell_\infty$-robust models on CIFAR-10. Results for other threat models and datasets can be found in App. F.

**Progress on adversarial defenses.** In Fig. 2, we plot a breakdown over conferences, the amount of robustness overestimation reported in the original papers, and we also visualize the robustness-accuracy trade-off for the $\ell_\infty$-models from the Model Zoo. First, we observe that for multiple *published* defenses, the reported robust accuracy is highly overestimated. We also find that the use of extra data is able to alleviate the robustness-accuracy trade-off as suggested in previous works [108]. However, so far all models with high robustness to perturbations of $\ell_\infty$-norm up to $\varepsilon = 8/255$ still suffer from noticeable degradation in clean accuracy compared to standardly trained models. Finally, it is interesting to note that the best entries of the $\ell_p$-leaderboards are still variants of PGD adversarial training [88, 168] but with various enhancements (extra data, early stopping, weight averaging).

**Performance across various distribution shifts.** We test the performance of the models from the Model Zoo on different distribution shifts ranging from common image corruptions (CIFAR-10-C, [58]) to dataset resampling bias (CIFAR-10.1, [112]) and image source shift (CINIC-10, [31]). For each of these datasets, we measure standard accuracy, and Fig. 3 shows that improvement in robust accuracy (which often comes with an improvement in standard accuracy) on CIFAR-10 correlates with an improvement in standard accuracy across distributional shifts. On CIFAR-10-C, robust models (particularly with respect to the $\ell_2$-norm) tend to give a significant improvement which agrees with the findings in [43]. Concurrently with our work, Taori et al. [137] study the robustness to different distribution shifts of many models trained on ImageNet, including some $\ell_p$-robust models. Our conclusions qualitatively agree with theirs, and we hope that our collected set of models will help to provide a more complete picture. Moreover, we measure robust accuracy, in the same threat model used on CIFAR-10, using AutoAttack [28] (see Fig. 10 in Appendix F), in order to see how $\ell_p$ adversarial robustness generalizes across the datasets representing different distributions shifts, and observe a clear positive correlation between robust accuracy on CIFAR-10 and its variations.

**Calibration.** A classifier is *calibrated* if its predicted probabilities correctly reflect the actual accuracy [52]. In the context of adversarial training, calibration was considered in Hendrycks et al. [61] who focus on improving accuracy on common corruptions and in Augustin et al. [7] who focus mostly on

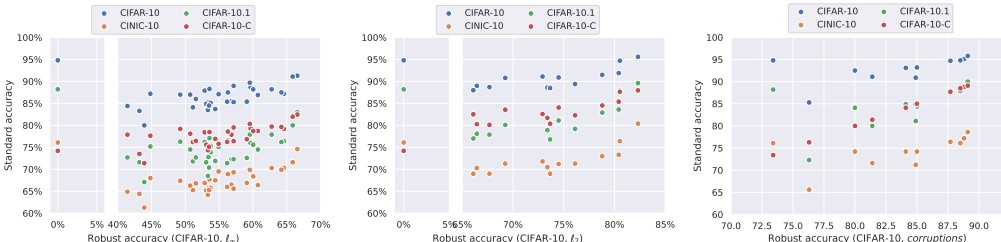

Figure 3: Standard accuracy of classifiers trained against $\ell_\infty$ (**left**), $\ell_2$ (**middle**), and common corruption (**right**) threat model respectively, from our Model Zoo on various distribution shifts.

preventing overconfident predictions on out-of-distribution inputs. We instead focus on *in-distribution* calibration, and in Fig. 4 plot the expected calibration error (ECE) without and with temperature rescaling [54] to minimize the ECE (which is a simple but effective post-hoc calibration method, see Appendix F for details) together with the optimal temperature for a large set of $\ell_\infty$ models. We observe that most of the $\ell_\infty$ robust models are significantly *underconfident* since the optimal calibration temperature is less than one for most models. The only two models in Fig. 4 which are *overconfident* are the standard model and the model of Ding et al. [34] that aims to maximize the margin. We see that temperature rescaling is even more important for robust models since without any rescaling the ECE is as high as 70% for the model of Pang et al. [101] (and 21% on average) compared to 4% for the standard model. Temperature rescaling significantly reduces the ECE gap between robust and standard models but it does not fix the problem completely which suggests that it is worth incorporating calibration techniques also during training of robust models. For $\ell_2$ robust models, the models can be on the contrary *more calibrated* by default, although the improvement vanishes if temperature rescaling is applied (see Appendix F).

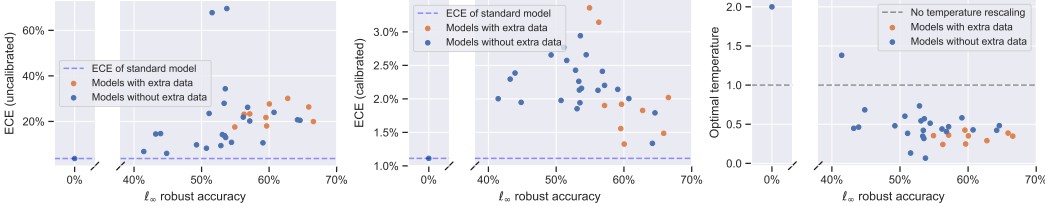

Figure 4: Expected calibration error (ECE) before (**left**) and after (**middle**) temperature rescaling, and the optimal rescaling temperature (**right**) for the $\ell_\infty$-robust models.

**Out-of-distribution detection.** Ideally, a classifier should exhibit high uncertainty in its predictions when evaluated on *out-of-distribution* (OOD) inputs. One of the most straightforward ways to extract this uncertainty information is to use some threshold on the predicted confidence where OOD inputs are expected to have low confidence from the model [59]. An emerging line of research aims at developing OOD detection methods in conjunction with adversarial robustness [57, 120, 7]. In particular, Song et al. [132] demonstrated that adversarial training [88] leads to degradation in the robustness against OOD data. We further test this observation on all $\ell_\infty$-models trained on CIFAR-10 from the Model Zoo on three OOD datasets: CIFAR-100 [75], SVHN [97], and Describable Textures Dataset [24]. We use the area under the ROC curve (AUROC) to measure the success in the detection of OOD data, and show the results in Fig. 5. With $\ell_\infty$ robust models, we find that compared to standard training, various robust training methods indeed lead to degradation of the OOD detection quality. While extra data in standard training can improve robustness against OOD inputs, it fails to provide similar improvements with robust training. We further find that $\ell_2$ robust models have in general comparable OOD detection performance to standard models (see Fig. 12 in Appendix), while the model of Augustin et al. [7] achieves even better performance since their approach explicitly optimizes both robust accuracy and worst-case OOD detection performance.

**Fairness in robustness.** Recent works [8, 160] have noticed that robust training [88, 168] can lead to models whose performance varies significantly across subgroups, e.g. defined by classes. We will refer to this performance difference as *fairness*, and here we study the influence of robust training

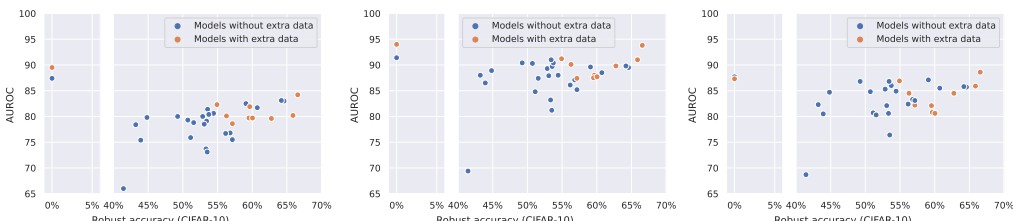

Figure 5: Visualization of the OOD detection quality (higher AUROC is better) for the $\ell_\infty$-robust models trained on CIFAR-10 on three OOD datasets: CIFAR-100 (**left**), SVHN (**middle**), Describable Textures (**right**). We detect OOD inputs based on the maximum predicted confidence [59].

methods on fairness. In Fig. 6 we show the breakdown of standard and robust accuracy for the $\ell_\infty$ robust models, where one can see how the achieved robustness largely varies over classes. While in general the classwise standard and robust accuracy correlate well, the class "deer" in $\ell_\infty$-threat model suffers a significant degradation, unlike what happens for $\ell_2$ (see Appendix F), which might indicate that the features of such class are particularly sensitive to $\ell_\infty$-bounded attacks. Moreover, we measure fairness with the relative standard deviation (RSD), defined as the standard deviation divided over the average, of robust accuracy over classes for which lower values mean more uniform distribution and higher robustness. We observe that better robust accuracy generally leads to lower RSD values which implies that the disparity among classes is reduced. (with a strong linear trend): improving the robustness of the models has then the effect of reducing the disparities among classes. However, some training techniques like MART [148] can noticeably increase the RSD and thus *increase the disparity* compared to other methods which achieve similar robustness (around 57%).

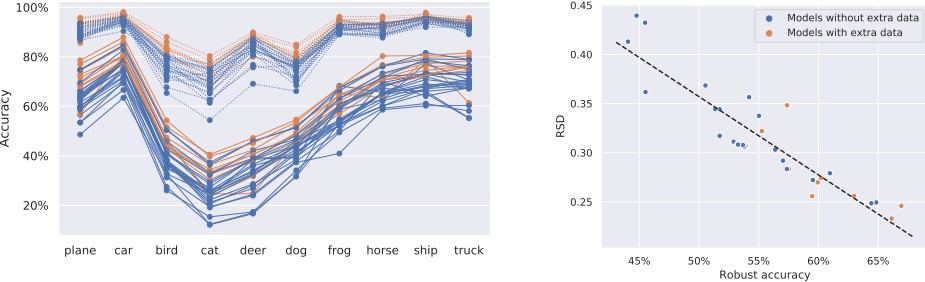

Figure 6: Fairness of $\ell_\infty$-robust models. **Left:** classwise standard (dotted lines) and robust (solid) accuracy. **Right:** relative standard deviation (RSD) of robust accuracy over classes vs its average.

**Privacy leakage.** Deep neural networks are prone to memorizing training data [127, 18]. Recent work has highlighted that robust training exacerbates this problem [131]. We benchmark privacy leakage of training data across robust networks (Fig. 7). We calculate membership inference accuracy using output confidence of adversarial images from the training and test sets (see Appendix F for more details). It measures how accurately we can infer whether a sample was present in the training dataset. Our analysis reveals mixed trends. First, our results show that not all robust models have a significantly higher privacy leakage than a standard model. We find that the inference accuracy across robust models has a large variation, where some models even have lower privacy leakage than a standard model, and there is no strong correlation with robust accuracy. In contrast, it is largely determined by the generalization gap, as using the classifier confidence does not lead to a much higher inference accuracy than the baseline determined by the generalization gap (as shown in Fig. 7 (right)). Thus one can expect lower privacy leakage in robust networks as previous work explicitly aimed to reduce the generalization gap in robust training e.g. via early stopping [113, 168, 50].

**Extra experiments.** In Appendix F, we show extra experiments related to the points analyzed above and describe some of the implementation details. Also, we study how adversarial perturbations transfer between different models. We find that adversarial examples strongly transfer from robust to robust, non-robust to robust, and non-robust to non-robust networks. However, we observe poor transferability of adversarial examples from robust to non-robust networks. Finally, since prior works [56, 162] connected higher smoothness with better robustness, we analyze the smoothness of the

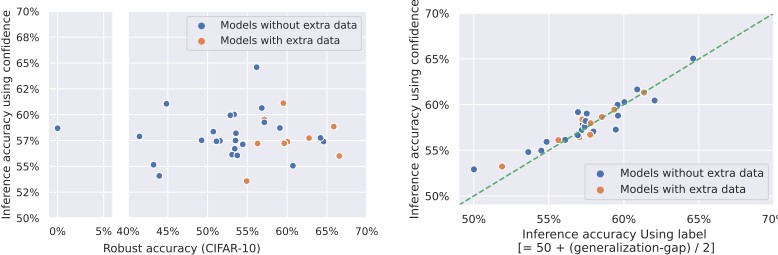

Figure 7: Privacy leakage of $\ell_\infty$-robust models. We measure privacy leakage of training data in robust networks and compare it with robust accuracy (**left**) and generalization gap (**right**).

models both at intermediate and output layers. This confirms that, for a fixed architecture, standard training yields classifiers that are significantly less smooth than robust ones. This study of properties of networks illustrates another useful aspect of our Model Zoo.

## 5  Outlook

**Conclusions.** We believe that a *standardized* benchmark with clearly defined threat models, restrictions on submitted models, and tight upper bounds on robust accuracy is useful to show which ideas in training robust models are the most successful. While AutoAttack is for most models very reliable and accurate and allows a standardized comparison, we ensure by flagging potentially unreliable evaluations and doing additional adaptive attacks that the benchmark reflects the best possible robustness assessment with limited resources as the exact robustness evaluation is computationally infeasible. We remark that recent works have already referred to our leaderboards [74, 163, 89, 136, 159], in particular as reflecting the current state of the art [111, 83, 103], and used the networks of our Model Zoo to test new adversarial attacks [92, 115, 41, 119], evaluate test-time defenses [146] or perceptual distances derived from them [67], explore further properties of robust models [134, 172]. Additionally, we have shown that unified access to a *large* and *up-to-date* set of robust models can be useful to analyze multiple aspects related to robustness. First, one can easily analyze the progress of adversarial defenses over time including the amount of robustness overestimation and the robustness-accuracy tradeoff. Second, one can conveniently study the impact of robustness on other performance metrics such as accuracy under distribution shifts, calibration, out-of-distribution detection, fairness, privacy leakage, smoothness, and transferability. Overall, we think that the community has to develop a better understanding of how different types of robustness affect other aspects of the model performance and `RobustBench` can help to achieve this goal. Finally, we note that a good performance on our benchmark does not guarantee the safety of the benchmarked model in a real-world deployment since $\ell_p$- and corruption robustness may not be sufficiently representative of all realistic threat models.

**Future plans.** Our intention in the future is to keep the current leaderboards up-to-date (see the maintenance plan in Appendix C) and add new leaderboards for other datasets and other threat models which become widely accepted in the community. We see as potential candidates (1) sparse perturbations, e.g. bounded by $\ell_0$, $\ell_1$-norm or adversarial patches [11, 26, 93, 29], (2) multiple $\ell_p$-norm perturbations [140, 90], (3) adversarially optimized common corruptions [68, 69], (4) a broad set of perturbations unseen during training [81]. Another possible direction is the development of a standardized evaluation of recent defenses based on some form of test-time adaptation [126, 146], which do not fulfill the third restriction (no optimization loop). Finally, although the benchmark currently focuses on image classification, we think that its structure and principles should apply to other tasks (e.g., image segmentation [157], image retrieval [139]) and domains (e.g., natural language processing [2], malware detection [51]) where adversarial robustness can be of interest. Since this direction requires more domain-specific expertise, we welcome contributions from different communities to expand RobustBench.

## Acknowledgements

We thank the authors who granted permission to use their models in our library. We also thank Chong Xiang for the helpful feedback on the benchmark, Eric Wong for the advice regarding the name of

the benchmark, and Evan Shelhamer for the helpful discussion on test-time defenses. Moreover, we thank the reviewers of both rounds of the NeurIPS 2021 Datasets and Benchmarks Track for their very useful suggestion that helped to improve the paper and make the discussion on standardized and adaptive attacks more balanced.

F.C. and M.H. acknowledge support from the German Federal Ministry of Education and Research (BMBF) through the Tübingen AI Center (FKZ: 01IS18039A), the DFG Cluster of Excellence "Machine Learning – New Perspectives for Science", EXC 2064/1, project number 390727645, and by DFG grant 389792660 as part of TRR 248. V.S. and P.M. acknowledge the support of the National Science Foundation (grants CNS-1553437 and CNS-1704105), the Army Research Office Young Investigator Prize, Army Research Laboratory (ARL) Army Artificial Intelligence Institute (A2I2), Office of Naval Research (ONR) Young Investigator Award, Schmidt DataX Fund, and Princeton E-ffiliates Partnership.

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
