# OpenReview forum: "RobustBench: a standardized adversarial robustness benchmark"
_NeurIPS.cc/2021/Track/Datasets_and_Benchmarks/Round2 — NeurIPS 2021 Datasets and Benchmarks Track (Round 2)_

### Official Review · Reviewer_RAKV · 2021-09-18
**A standardized benchmark on evaluating adversarial robustness in image classification**

**Rating:** 7
**Confidence:** 5
**Clarity:** The paper is well written and easy to…

**Strengths:**

* The benchmark provides a leaderboard with more than 120 evaluations that help to track the progress of the domain.
* A model zoo is built to facilitate the usage of SOTA models.
* Thorough analyses are provided to see the impact of different aspects on the adversarial robustness.
* The authors address the concerns of reviewers in Round 1 and add clarification in the paper.

**Weaknesses:**

* The paper is titled as a standardized adversarial robustness benchmark, however, it actually only focuses on the very specific task of image classification.
* For all figures in Section 4, it is impossible to identify the specific methods, which makes it hard to find the most effective method.
* Currently, three datasets are provided (CIFAR-10, CIFAR-100, ImageNet). If the ranking differs in different datasets, how to evaluate the adversarial robustness of the method? Furthermore, are there any other contradictory results found using the benchmark (e.g. ranking changes when varying the attack constraints)? In this case, how can one determine a model is more robust than another?

**Additional Feedback:**

My concerns are listed in weakness. I think the benchmark is an interesting work to the community and I hope the authors keep the promise to regularly update it.

**Correctness:**

The benchmark is established in a reasonable way. The evaluation results in experiments seem correct.

**Documentation:**

The benchmark is well-documented and reproducibility should be guaranteed by the open-source code.

**Ethics:**

No ethical concerns.

**Relation To Prior Work:**

The work is compared with prior works. Works like FoolBox, Cleverhans are toolkits rather than benchmarks. Benchmark like RobustML was designed differently and is not updated for a long time.

**Summary And Contributions:**

Robustness overestimation is a major problem in the research community of adversarial robustness. The paper proposes RobustBench, a standardized benchmark that addresses this problem. With AutoAttack, an ensemble of white- and black-box attacks, the benchmark improves the evaluation in most of the previous works. A leaderboard is provided with more than 120 models that help to track the progress of the domain. Extensive experiments with RobustBench show various aspects of robust classifiers, including calibration, OOD detection, privacy leakage, etc. Overall, the proposed benchmark can help the community to better understand the adversarial robustness of image classification.

---

> ### Author Response · Authors · 2021-09-29
> **Response to Reviewer RAKV**
>
> We thank Reviewer RAKV for the positive feedback on our paper. We address the raised points below.\
> \
> **"The paper is titled as a standardized adversarial robustness benchmark, however, it actually only focuses on the very specific task of image classification"**
>
> We agree that currently the focus on image classification tasks is not clearly stated. We update the abstract to clarify this status, but we leave the title more general since we hope that the benchmark will grow, with the help of the community, to include new tasks and domains.\
> \
> **"For all figures in Section 4, it is impossible to identify the specific methods, which makes it hard to find the most effective method"**
>
> The main goal of the figures is to show the correlation between robustness and the other metrics of interest, rather than ranking the individual techniques. We also note that in most cases, it is possible to identify the exact model by its robust accuracy, although this is not the goal of this section. In addition to the leaderboard on the webpage, we also provide a list of all models in the leaderboard in Appendix of the paper (Tables 3-9).\
> \
> **"Currently, three datasets are provided (CIFAR-10, CIFAR-100, ImageNet). If the ranking differs in different datasets, how to evaluate the adversarial robustness of the method?"**
>
> Our benchmark focuses on ranking particular **models** (rather than particular **methods**) which have been trained on specific datasets under specific threat models. It is possible that the ranking between different **methods** varies over different datasets, model architectures or other factors. This can lead to an interesting meta analysis and we leave it to the users of the benchmark.\
> \
> **"Furthermore, are there any other contradictory results found using the benchmark (e.g. ranking changes when varying the attack constraints)? In this case, how can one determine a model is more robust than another?"**
>
> We clarify that, for each defense, we test a classifier only in the threat model to which it was trained to be robust (with the exception of the most $\ell_p$-robust models which we use as baselines for the common corruption leaderboard). It is possible that some defenses are more specific and effective for a certain threat model than others, although currently the best performing methods are roughly consistent across types of perturbations and datasets. Moreover, we note that our benchmark allows such comparisons to better understand the differences among defenses.

---

> > ### Comment · Reviewer_RAKV · 2021-09-29
> > **Response to the authors**
> >
> > Thank you for the explanations. I'd like to support the acceptance of this work.

---

### Official Review · Reviewer_YCwj · 2021-09-20

**Rating:** 7
**Confidence:** 4
**Clarity:** Yes, the paper is well written.

**Strengths:**

1. It is very important to have a standard adversarial benchmark and leaderboard for the community.

2. The paper has included most of the important datasets and models for a comprehensive evaluation.

3. It also provided a deep and interesting analysis of the benchmark.

**Weaknesses:**

1. The paper missed a few recent works on the adversarial attack and library, for example:
a. RayS: A Ray Searching Method for Hard-label Adversarial Attack
b. DeepRobust: A PyTorch Library for Adversarial Attacks and Defenses

**Additional Feedback:**

My overall evaluation of the paper is good. The author can address the minor comments I made about the related works.

**Correctness:**

The paper provided enough details of how the dataset and the model zoo are collected.

**Documentation:**

The documentation as well as the website of the leaderboard, provide a clear description of the dataset, submission and evaluation.

**Ethics:**

There is no ethical concern.

**Relation To Prior Work:**

The paper included most of the prior works related to adversarial benchmarks. But still, it missed a few important ones.

**Summary And Contributions:**

This paper provides a standard adversarial benchmark for evaluating adversarial training methods. By collecting 120+ models on different datasets, it builds a leaderboard based on AutoAttack. In addition, it also analyzes the impact of robustness on the performance on distribution shifts, calibration, out-of-distribution detection, etc.

---

> ### Author Response · Authors · 2021-09-29
> **Response to Reviewer YCwj**
>
> We thank Reviewer YCwj for the positive assessment of our paper. We comment on the suggestion about related works below.
>
> **"The paper missed a few recent works on the adversarial attack and library, for example: a. RayS: A Ray Searching Method for Hard-label Adversarial Attack b. DeepRobust: A PyTorch Library for Adversarial Attacks and Defenses"**
>
> Thanks for mentioning these relevant papers, we have added them to the paragraph **“Related libraries and benchmarks”**. However, we note that both have different focus than ours: [a] proposes and tests a new hard-label attack on several defenses, while [b] provides implementations of many commonly used attacks and defenses, without introducing a benchmark or a model zoo.\
> \
> We would be happy to answer any further questions and discuss suggestions that you may have.

---

### Official Review · Reviewer_nV8d · 2021-09-21
**"RobustBench: a standardized adversarial robustness benchmark" - a strong and carefully designed contribution for advancing research in adversarial robustness**

**Rating:** 7
**Confidence:** 3

**Strengths:**

- This work represents a valuable contribution to the ongoing and very active field of adversarial machine learning, and especially to the research into adversarial robustness.
- The associated  - currently continuosly updated and curated -  leaderbords of many published models and the robust model zoo together with the very accessible open source implementation and easily usable wrapping into a python package represents extremely valuable, commendable and probably also sometimes tedious work. All the more so, when one additionally considers the efforts to reach out into the community for contributions and corrections.
- While the presented limitations and restrictions can seem quite extensive and thus render the direct usability of the benchmark less general and more specifically geared towards image classification problems under  threat models of $l_{inf}$ and $l_{2}$ (with additional restrictions on the listed defenses and no standard exploration of adaptively estimated adversarial robustness, and the almost complete exlusion of verified robustness methods etc.), it is absolutely necessary that this type of work is conducted to advance the field. Thus, I agree with the basic assessment that a standardized benchmark needs to start with a set of specific (standardizable) problems, and the choices seem mostly reasonable to me, i.e. using the most studied case of image classification together with the most widely studied threat models. Of course, this is only true as long as all limitations are clearly stated, which too a large degree is done by the authors (see below for minor criticism).
- With section 4, the study is already providing a range of analyses based on the benchmarks regarding the impact of adversarial robustness in models on a range of subjects such as distributional shifts, out-of-distribution detection, privacy leakage, transferability, especially when considering the rich supplementary material. This work already provides an example of the wide range of analysis that is possible with this kind of benchmark and even draws many smaller conclusions with regard to existing literature.

**Weaknesses:**

Apart from the  stated limitations of the benchmark themselves (which i don't necessarily consider as weaknesses) there are two main points of criticism:
- The limitations are not always fully laid out to the reader in a clear way. This is certainly in part due to the complexity of the discussed problem and the limited space, where some limitations just are being stated shortly at a later stage when the necessary background is clear (such as which data sets are used, or which exact model architectures are benchmarked). However,  the title of the publication seems too broad and also the abstract does not mention some of the limitations such as  the considered threat models, how adaptive attacks are considered, the restriction to mostly empirical adversarial robustness and upper bounds, or the restriction to image classification problems. I would encourage the authors to stand more clearly by their own words in the  supplemental materials, section A, where in a rebuttal to a previous reviewer they made the clear point that even creating such benchmark just for the most studied problem of image classification and $l_{p}$ robustness is already very challenging. Indeed!
- The rich supplementary material is both a good and a problematic element. Clearly, the authors have a lot more to present and say than what they could reasonably provide within 9 pages and much of it is interesting and valuable. The specific choice is sometimes a bit unfortunate, because some of the material would enrich the main text, which sometimes borders to a point where the supplementary material actually becomes necessary to fully understand (or at least appreciate) the main text. A clear example is line 64, in which the contributions from further analysis are listed and among them are several topics only addressed in detail in the supplementary material (fairness, transferability).

A few additional smaller points
- it is unclear to me, why the tested models seem to be exclusively based on the Resnet and wideResnet architecture (line 255). Is there a specifc reason for this? And shouldn't that be more openly considered a bias?
- In 257-262 it is elaborated that some defenses of additional goals alongside pure adversarial robustness and that these factors need to be taken into account. However, a discussion is missing as to whether RobustBench is doing this (or not, and then, why).
- clearly, the AutoAttack package is a cornerstone of RobustBench. I would have liked more details on the package and how its ensemble of attacks was designed. In particular, it seems important how the standardized robust accuracy cited in the leaderboards of RobustBench has been derived from the ensemble of attacks. As it stands, i have the feeling that it is almost necessary to also read the publication of the AutoAttack package to get the full view on RobustBench.



**Additional Feedback:**

The authors have already published an article of the same title in the ICLR 2021 workshop on Security and Safety in Machine Learning Systems, see here https://aisecure-workshop.github.io/aml-iclr2021/papers/47.pdf

There is significant overlap between the two versions, albeit the current version is substantially fruther developed. I did not consider this in my review (also because i am not fully aware of the details of Neurips policy on this kind of situation) so far and i'd like to give the authors the room to justify  their new submission to this conference, if necessary at all (e.g. it may already have been cleared).

**Clarity:**

The paper is well written, clear and understandable. It seems to me that there are a few smaller issues with image captions:
- line 329 captions in figure 6 seem wrong (three different data sets)
- line 313 captions in figure 4 are maybe wrong (all l_inf?)

**Correctness:**

To the best of my knowledge all the presented work seems to be correct. The study is deeply grounded in existing literature and practices with many relevant references throughout the pages. The benchmark is constructed in a sound way, with much attention to detail, limitations, safeguards (e.g. the flag in AutoAttack and subsequent individual testing) and cross-referencing other methods and contributions from the community, if available.

**Documentation:**

The paper contains all important links to the RobustBench webpage and github repository. It was very easy to install robustbench with the pip package manager and directly explore the benchmark and model zoo. It seems easy to integrate the benchmark into your own code and projects. There is extensive documentation available on the webpages, all code is open sourced. Lincenses are properly discussed in the supplementary material.

**Ethics:**

No special concern beyond the larger ongoing discussions around research into adversarial machine learning. But even there, this paper rather contributes positively, since it is concerned with creating more strict and standardized ways of assesing the robustness of models.

**Relation To Prior Work:**

 The paper contains an extensive list of references and is deeply embedded in the ongoing research work on adversarial robustness. There is a specific discussion of the differences and advantages of RobustBench in relation to other adversarial robustness benchmark plattforms (RobustML, Deepsec).

**Summary And Contributions:**

This submission presents a standardized benchmark for empirical adversarial robustness under a set of specific restrictions in order to keep the benchmark and associated leaderbords and model zoo more meaningful, useful and standardized at the (openly discussed) expense of a broader coverage of attack types, defenses, adversarial robustness measures and application settings.  The benchmark consists of a currently continously updated set of leaderbords for adversarial robustness and a model zoo of the tested robust models, all hosted on a website and github. The benchmark exlusively focuses on the problem of adversarial robustness in the context of image classification models under  threat models of $l_{inf}$ and $l_{2}$ perturbations represented by the attacks implemented in the autoattack package, allowing for certain set of defenses in the benchmarked robust models.

---

> ### Author Response · Authors · 2021-09-29
> **Response to Reviewer nV8d**
>
> We thank Reviewer nV8d for the positive feedback on our paper. We address the mentioned concerns below.\
> \
> **"The limitations are not always fully laid out to the reader in a clear way… the title of the publication seems too broad and also the abstract does not mention some of the limitations such as  the considered threat models, how adaptive attacks are considered, the restriction to mostly empirical adversarial robustness and upper bounds, or the restriction to image classification problems…"**
>
> We would like to clarify that we do mention in the abstract the exact threat models (*“well-defined tasks in Linf- and L2-threat models and on common corruptions”*) and imply that adaptive evaluations will be taken into account (*“To prevent overadaptation of new defenses to AutoAttack, we welcome external evaluations based on adaptive attacks [137], especially where AutoAttack flags a potential overestimation of robustness.”*). Moreover, we say that we rely on AutoAttack to measure robustness which (as any other empirical attack) provides only upper bounds.\
> However, we indeed do not mention that so far the benchmark focuses only on the image classification task which is a great suggestion which we incorporate.\
> \
> **"the supplementary material actually becomes necessary to fully understand (or at least appreciate) the main text. A clear example is line 64, in which the contributions from further analysis are listed and among them are several topics only addressed in detail in the supplementary material (fairness, transferability)."**
>
> Indeed, we lacked the space to present all the results mentioned in line 64 so we had to select the most interesting ones for the main part. With the additional page allowed in the camera-ready version, we are able to partially expand Section 4 (now including the analysis about fairness). However, the important idea behind the analysis section is not only presenting the specific findings (that can be found also in appendix) but also highlighting the benefits of having a large Model Zoo for very different kinds of analyses. In fact, we expect that other researchers can find the Model Zoo useful for many other applications that go beyond what we have shown in the paper.\
> \
> **"the tested models seem to be exclusively based on the Resnet and wideResnet architecture (line 255). Is there a specifc reason for this? And shouldn't that be more openly considered a bias?"**
>
> ResNets are currently the most popular backbones in the literature on adversarial robustness, so this fact is just reflected in our benchmark. We do not consider it a bias as we are completely open to any architecture. We expect that this may change in the future with the development of transformer models.\
> \
> **"In 257-262 it is elaborated that some defenses of additional goals alongside pure adversarial robustness and that these factors need to be taken into account. However, a discussion is missing as to whether RobustBench is doing this (or not, and then, why)"**
>
> We highlight such cases (fast adversarial training, compressed models) in the footnotes for the corresponding entries in the leaderboard.\
> \
> **"more details on the package and how its ensemble of attacks was designed. In particular, it seems important how the standardized robust accuracy cited in the leaderboards of RobustBench has been derived from the ensemble of attacks."**
>
> We provide a description of AutoAttack in L196-200: *”To evaluate robustness ... we currently use AutoAttack [26]. It is an ensemble of four attacks that are run sequentially: a variation of PGD attack with automatically adjusted step sizes, with (1) the cross entropy loss and (2) the difference of logits ratio loss, which is a rescaling-invariant margin-based loss function, (3) the targeted version of the FAB attack [25], which minimizes the Lp-norm of the perturbations, and (4) the black-box Square Attack [3].”*\
> However, we agree that the description can be improved by mentioning that the attacks are not only run sequentially but that each subsequent attack is run on samples for which an adversarial example has not been found by the preceding attacks. The final robust accuracy is the percentage of samples for which no adversarial example is found after all four attacks. We have updated the text of the paper accordingly.\
> \
> **"The authors have already published an article of the same title in the ICLR 2021 workshop on Security and Safety in Machine Learning Systems"**
>
> We note that “The workshop is non-archival and will not have any official proceedings” (https://aisecure-workshop.github.io/aml-iclr2021/cfp) and according to https://neurips.cc/Conferences/2021/CallForPapers “Work that has appeared in non-archival workshops, such as workshops at NeurIPS/ICML, may be submitted”. Also, we have significantly expanded the paper with new models, leaderboards (e.g., for ImageNet), analyses, and added a flag in AutoAttack that can inform the user about potential robustness overestimation.

---

> > ### Comment · Reviewer_nV8d · 2021-09-29
> > **Response to the authors**
> >
> > Thank you for the detailed response and the feedback reflected in the updated document. I see most of my leftover points addressed and also have to apologize for clearly not reading the abstract carefully enough since indeed more points were already addressed than i had claimed.
> >
> > Were I still somewhat disagree (without any strong opinion that this should be addressed any further, more for the discourse here) are
> >
> > - that the authors want to keep the title general (as explained in the response below to reviewer RAKV) in the hope to enlarge the benchmark in future. While i strongly share that hope, and surely agree that, for instance, the website could easily keep the general title, i still think that for this particular publication, a more clear title reflecting the content at this point in time would be better.
> >
> > - i agree that the model zoo is important to be discussed and that the analysis section also serves as some sort of advertising of its strengths. However, my point was that in the original paper (and still maybe to a very minor degree), what was summarized as a contribution from the analysis section only could be found when also considering the supplemental materials. I don't mind at all to have a lot of elements in the supplemental materials, but in my opinion they should not constitute a part of the claimed contributions of the paper itself.
> >
> > Thanks again for the response and congratulations for the great paper and work!

---

### Official Review · Reviewer_vjuH · 2021-09-21
**Well-designed benchmark that fills a need**

**Rating:** 8
**Confidence:** 4

**Strengths:**

Given that there are over 3000 papers written on the topic of adversarial examples, with a large subset of these papers proposing new techniques to train robust models and defend against adversarial examples, the community is in need of a systematic way to track and compare these defenses. RobustBench is perhaps the best attempt to date to do this, overcoming the limitations of prior work like robust-ml.org that attempted to solve this problem.

RobustBench makes several important insights. One is that many reasonable defenses satisfy a reasonable set of restrictions: (1) differentiable, (2) deterministic, and (3) free of optimization loops. The other is that AutoAttack, a parameter-free attack that performs variations of several well-known white-box and black-box attacks, often gives reasonably good upper bounds for defenses that satisfy that set of restrictions.

Basing a benchmark/leaderboard on AutoAttack allows for reasonable comparison between defenses even when the defenses haven't been carefully analyzed by third-party researchers seeking to perform an adaptive attack, which makes the benchmark much more useful overall. Adding a new defense to the benchmark is easy and useful. RobustBench recognizes that AutoAttack isn't perfect, and so it supports submissions of adaptive attacks on specific defenses as well.

RobustBench already appears to have a demonstrated usefulness and traction among the community.


**Weaknesses:**

The single biggest concern I have with RobustBench's approach is that the popularity/acceptance of RobustBench as _the_ way to evaluate adversarial robustness may lead to future work on adversarial examples de-emphasizing adaptive attacks and "overfitting" to AutoAttack. I know that RobustBench does accept adaptive attacks, but do the authors have ideas on whether greater emphasis should be placed on those, and if so, how? Could the leaderboard website better emphasize adaptive attacks? A related question: what is the default sort order for the leaderboard --- is it by AutoAttack accuracy, or best known robust accuracy?

Another conern I have is with the restrictions: how can the restrictions be enforced? Some should be possible to check in a straightforward way, e.g. ensuring that the model does not use randomness, but how will the property of non-zero gradients be enforced? It should be easy to catch models that are "obviously" non-differentiable, e.g. models using quantization, but what about models that are technically differentiable but often end up having "non-useful" gradients, e.g. due to numerical instability? Such models could "break" AutoAttack without actually being robust.

Minor concerns:

- The abstract mentions "reasonable computational budget". What does this mean? Is it even necessary to include as part of the threat model? The threat model already bounds perturbation distance under an Lp norm.
- How reliable is the "flag" added to AutoAttack? For defenses that satisfy the three restrictions, how often is it the case that an adaptive attack performs significantly better than AutoAttack, and in how many of those cases did AutoAttack raise a flag? Is there a danger of the flag being misleading with false negatives?
- Why does the benchmark allow submitting a defense without publicly sharing the checkpoint? It seems like that would make it hard/impossible for other researchers to perform adaptive attacks on the defense.


**Additional Feedback:**

- I found the orange text in Figure 1 and the website a bit difficult to read. (The contrast ratio is below the value recommended by WCAG: https://web.dev/color-and-contrast-accessibility/)
- The model zoo seems very useful. Is there any harm in making it a requirement to integrate a model into the model zoo as part of submitting a new defense for inclusion in the leaderboard?


**Clarity:**

Yes; the paper as well as the website and GitHub issue templates are well-done.


**Correctness:**

Yes. The benchmark design seems like the best possible while having a fully automated attack. The paper has a careful treatment of threat models, empirical evaluations being upper bounds, and adaptive attacks.


**Documentation:**

Yes.

**Relation To Prior Work:**

The paper has a good comparison to prior work on libraries for attacks, as well as prior works on leaderboards/benchmarking of defenses.


**Summary And Contributions:**

This paper contributes RobustBench, a benchmark for adversarial robustness. The benchmark uses a standardized parameter-free attack called AutoAttack to give an upper bound for robustness, which obviates the need for third parties to perform time-consuming adaptive evaluations per-defense. For defenses that follow a set of restrictions, AutoAttack generally gives reasonable bounds. RobustBench does not currently support defenses that violate these restrictions. In case AutoAttack doesn't give a tight bound on a particular defense, RobustBench allows third parties to submit results from an adaptive attack.

In addition to RobustBench, the paper contributes a model zoo for adversarially robust models, giving researchers a unified interface to over 80 robust models proposed in the literature. Finally, the paper gives some analyses of robust models that are enabled by the unified API of the model zoo.

---

> ### Author Response · Authors · 2021-09-29
> **Response to Reviewer vjuH - part 1**
>
> We thank Reviewer vjuH for the positive assessment of our paper. We address the raised concerns below.\
> \
> **"The single biggest concern I have with RobustBench's approach is that the popularity/acceptance of RobustBench as *the* way to evaluate adversarial robustness may lead to future work on adversarial examples de-emphasizing adaptive attacks and "overfitting" to AutoAttack. I know that RobustBench does accept adaptive attacks, but do the authors have ideas on whether greater emphasis should be placed on those, and if so, how? Could the leaderboard website better emphasize adaptive attacks?"**
>
> This is a great point and we do not want to imply that AutoAttack should become **the** only tool for robustness evaluation. To reflect this idea, right in the abstract we mention the possibility of potential robustness overestimation of AutoAttack and point to adaptive attacks as complementary to AutoAttack (Lines 12-14): *“To prevent overadaptation of new defenses to AutoAttack, we welcome external evaluations based on adaptive attacks [137], especially where AutoAttack flags a potential overestimation of robustness.”*
> We also reiterate this point in the main text, e.g., in paragraph **“Identifying potential need for adaptive attacks”** (Line 218) where we describe a case where AutoAttack overestimates the robustness which led to a new flag in AutoAttack which is triggered if Square Attack is able to non-trivially reduce the robust accuracy compared to the previous white-box attacks in the ensemble. Similarly when an adaptive attack performs better than AutoAttack, we also report its robust accuracy on the leaderboard.
> However, emphasizing the point about adaptive attacks more on the website and in the README on github is a great suggestion and we have updated them accordingly. We hope that this will help to make it clearer for the users of the benchmark that we consider standardized and adaptive attacks as *complementary* to each other, and that the standardized leaderboard can help facilitate adaptive attacks by focusing the community’s attention on the most promising models.\
> \
> **"A related question: what is the default sort order for the leaderboard --- is it by AutoAttack accuracy, or best known robust accuracy?"**
>
> The leaderboards are by default sorted by **the best known robust accuracy** which aims to prioritize the best known robustness evaluation (including adaptive attacks if they have been studied for a certain model). However, we note that the leaderboard can be sorted by any individual or multiple columns at once.\
> \
> **"Another conern I have is with the restrictions: how can the restrictions be enforced? Some should be possible to check in a straightforward way, e.g. ensuring that the model does not use randomness, but how will the property of non-zero gradients be enforced? It should be easy to catch models that are "obviously" non-differentiable, e.g. models using quantization, but what about models that are technically differentiable but often end up having "non-useful" gradients, e.g. due to numerical instability? Such models could "break" AutoAttack without actually being robust"**
>
> Indeed, in most of the cases, we can check the restrictions automatically for a given model. To facilitate this, we made a basic set of checks (or flags) *automatic* when running AutoAttack (see https://github.com/fra31/auto-attack/blob/master/flags_doc.md for a detailed description and exact criteria). However, there still can be cases when we may need to check the model implementation manually. For example, it is possible to redefine the backward pass of a quantization operation with its differentiable approximation (or even unrelated function) which would make a non-differentiable model output some non-zero gradients. This will not be captured by the automatic flags, thus we leave the possibility for checking the implementation manually.\
> Regarding the concern about differentiable models with “non-useful” gradients, we do not aim at excluding *all* such models since “non-useful” gradients are hard to clearly formalize except *particular obvious cases* which have led to our restrictions. We think they cover most of the cases that have been discussed in the literature but there can exist exceptions. One prominent known exception is the [k-Winners-Take-All](https://arxiv.org/abs/1905.10510) approach that relies on a discontinuous activation function which we describe in paragraph **“Identifying potential need for adaptive attacks”** (Line 218). However, to make sure that we adequately address such exceptions, we introduced a new flag in AutoAttack which is triggered if Square Attack is able to non-trivially reduce the robust accuracy compared to the previous white-box attacks in the ensemble. In this way, such exceptions that rely on “non-useful” gradients should be flagged in the leaderboard suggesting that adaptive attacks should be tried on these models,

---

> > ### Comment · Reviewer_vjuH · 2021-09-29
> > **x**
> >
> > Thank you for the response! Yes, on the topic of adaptive attacks, I think the paper did a great job; thank you for making this more clear in the website/GitHub as well. On the topic of restrictions: that is neat, I didn't realize that AutoAttack does some checks for restrictions automatically. That could be nice to mention in the paper.

---

> ### Author Response · Authors · 2021-09-29
> **Response to Reviewer vjuH - part 2**
>
> **"The abstract mentions "reasonable computational budget". What does this mean? Is it even necessary to include as part of the threat model? The threat model already bounds perturbation distance under an Lp norm."**
>
> The main rationale behind this statement was to clarify that we do not compare to the exact combinatorial solvers (e.g. [Evaluating Robustness of Neural Networks with Mixed Integer Programming](https://arxiv.org/abs/1711.07356)) which are able to compute the exact robust accuracy but do not scale to state-of-the-art models and thus require an infeasible amount of time to be run. Instead we use AutoAttack, which provides a reliable estimation of robust accuracy and easily scales up to state-of-the-art models. \
> Moreover, we would like to clarify that we do not consider the computational budget as a part of the threat model. Rather, we consider it as an important factor to take into account when selecting a standardized attack in practice. In particular, by a “reasonable computational budget” we mean a budget available to a standard academic lab so that it can reproduce a certain robustness evaluation within at most a few days. We refer to Appendix F where we show the time needed to run AutoAttack on a few representative models using a single V100 GPU.\
> \
> **"How reliable is the "flag" added to AutoAttack? For defenses that satisfy the three restrictions, how often is it the case that an adaptive attack performs significantly better than AutoAttack, and in how many of those cases did AutoAttack raise a flag? Is there a danger of the flag being misleading with false negatives?"**
>
> Currently, only one model among those on the leaderboards triggered the flag ([k-Winners-Take-All approach](https://arxiv.org/abs/1905.10510)), and that is the only case for which adaptive attacks substantially improve the evaluation of AutoAttack. In the other cases, where external evaluations are available, the difference in robust accuracy is at most 0.06%. At the moment we haven’t seen false negatives occur.\
> \
> **"Why does the benchmark allow submitting a defense without publicly sharing the checkpoint? It seems like that would make it hard/impossible for other researchers to perform adaptive attacks on the defense"**
>
> We allow this to keep the leaderboard up-to-date even with those defenses for which the code and checkpoints cannot be released immediately out of licencing reasons. This happened a few times in the past for some models coming from industrial research labs which were publicly released after some delay. \
> \
> **"I found the orange text in Figure 1 and the website a bit difficult to read. (The contrast ratio is below the value recommended by WCAG: https://web.dev/color-and-contrast-accessibility/)"**
>
> We thank you for this suggestion. We have changed the color of the links on the website to improve its accessibility and user experience.\
> \
> **"The model zoo seems very useful. Is there any harm in making it a requirement to integrate a model into the model zoo as part of submitting a new defense for inclusion in the leaderboard?"**
>
> For each submitted entry on the leaderboard, we encourage authors to also provide the corresponding model checkpoints. However, we also consider the possibility that some authors may prefer to distribute their models via their own repositories or libraries (e.g., due to specific licences), so we decided to leave this possibility open. However, this situation seems to be quite rare and nearly all the authors that we have contacted so far have agreed to provide their models to the Model Zoo.

---

### Decision · Program_Chairs · 2021-10-09

**Decision:**

Accept

**Comment:**

In this paper the authors provide a benchmark for adversarial robustness, enabling testing with a wide variety of attacks, nuanced discussion of incorporating adaptive attacks, and already evaluations and open-sourcing of numerous models.

The largest concern raised by multiple reviewers was on the limitations of this benchmark. As with most benchmarks, the discussion of the limitations of this leaderboard is crucial so as to not discourage new research outside the scope of focus of this framework. I'd strongly encourage the authors to continue to clarify and qualify the limited scope of this evaluation, e.g. in how the work is titled, on the website, etc.

That said, all reviewers agreed this was a very valuable resource to the community and as such I believe it should be accepted.